# AT-DRIVE: EXPLOITING ADVERSARIAL TRANSFER FOR END-TO-END AUTONOMOUS DRIVING

## ABSTRACT

End-to-end autonomous driving methods aim to enable robust vehicle control by imitating successful driving behavior. Existing approaches are trained either on real-world data, which closely reflects practical applications, or on simulation data, which is used to simulate undesirable behaviors such as non-compliant driving habits, car accidents, and off-road scenarios. However, these methods fail to integrate the advantages of both data sources effectively. In this paper, we propose AT-Drive, an end-to-end adversarial transfer framework for autonomous driving. AT-Drive is the first approach that transfer the simulated imitation driving capabilities to real-world deployment. AT-Drive first pre-trains simulation and real-world model with simulation and real-world dataset separately. Then, two discriminators are utilized to adversarially train the real-world model, producing a model that transfers simulation-based driving capabilities into real-world deployment. This approach bridges the gap between simulation and real-world autonomous driving. Furthermore, by incorporating a unique back-propagation strategy, AT-Drive achieves state-of-the-art performance on the newly partitioned nuScenes dataset.

## 1 INTRODUCTION

Autonomous driving is a complex system that requires not only a detailed representation of the environment, detecting both dynamic objects and static elements, but also trajectory prediction, navigation, path planning, and collision avoidance. Multi-stage autonomous driving algorithms struggle to handle various hard cases across different industrial scenarios. These multi-stage autonomous driving methods are typically composed of a series of independent modules for detection, tracking, online mapping, prediction, and planning. To unify these complex models and mitigate error accumulation and feature misalignment, researchers have developed a more streamlined and efficient pipeline that integrates all tasks into a single model. These approaches are co-trained on real-world or simulated data to evaluate their performance.

Some researchers train and evaluate their end-to-end models on real-world datasets for industrial applications. For example, Pomerleau (1988) uses a three-layer neural network trained on real camera data to predict the direction of the vehicle. Prakash et al. (2021) directly predicts planning trajectories without modeling complex interactions. UniAD Hu et al. (2023) introduces a unified framework that leverages the joint cooperation of separate tasks and complex interactions to improve performance. Additionally, Sun et al. (2024); Zhang et al. (2024) simplify the internal tasks with a parallel design to better align with real-world autonomous driving scenarios. Note that, models trained on real-world data often lack exposure to critical hard cases and corner cases, such as extreme weather, non-compliant driving behavior, and traffic accidents. To improve the robustness of the model in handling such cases, some researchers train and evaluate their models on simulated datasets. Specifically, Renz et al. (2022); Jia et al. (2023b) adopt transformers as their primary modules, demonstrating excellent performance in closed-loop evaluations. Furthermore, other approaches, such as DriveAdapterJia et al. (2023a), introduce a teacher-student method to effectively exploit the complex simulation scenarios.

Although simulations are well-suited for modeling real-world autonomous driving rare and challenging scenarios and can generate a large number of such scenarios, transferring a model trained in simulation to real-world applications remains challenging.

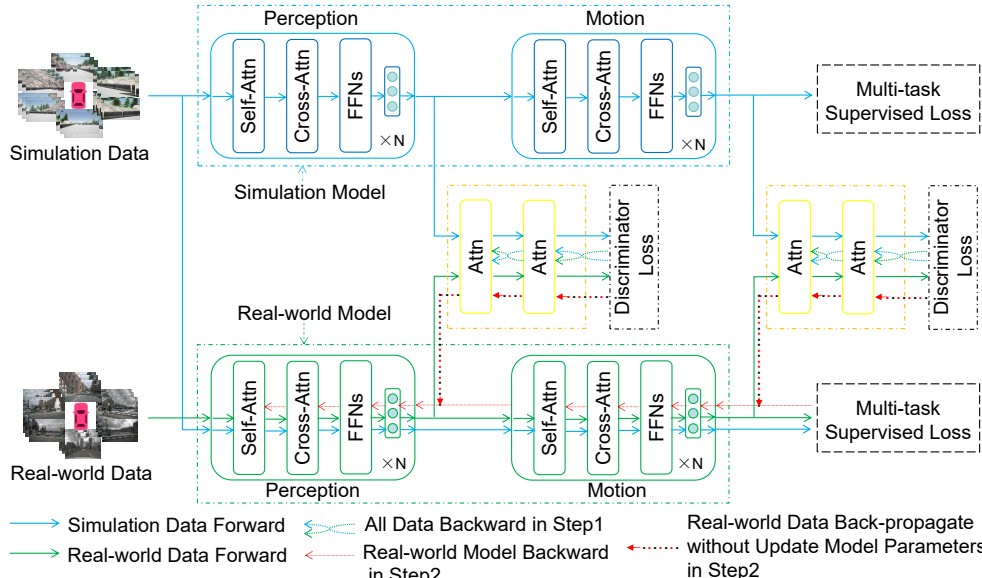

Figure 1: Overview of our proposed AT-Drive. AT-Drive is composed by a simulation end-to-end AD baseline, a real-world end-to-end AD baseline, and two discriminators. Simulation and real-world baselines are same model that trained on simulation dataset and real-world dataset respectively. Two discriminators are used to adversarial transfer the ability of simulation baseline to real-world baseline.

In this paper, we introduce a novel end-to-end autonomous driving method, named **AT-Drive**, which can effectively transfer the ability to handle simulation-based rare and challenging cases to real-world applications. The architecture is illustrated in fig. 1. AT-Drive consists of two base models (a simulation model and a real-world model) and two discriminators (perception discriminator and motion discriminator). The simulation model learns uncommon scenarios from synthetic data and transfers this knowledge to the real-world model through adversarial learning with the discriminators.

To the best of our knowledge, AT-Drive is the first approach to investigate and transfer simulation-trained capabilities to real-world autonomous driving. Furthermore, we reproduce several existing methods and compare them with AT-Drive on a newly partitioned dataset. Experimental results confirm the superior performance of our method, surpassing state-of-the-art approaches.

The contributions of this paper are as follows:

- We present the first end-to-end autonomous driving model that transfers simulation-based hard case handling ability to real-world scenarios, effectively bridging the domain gap between simulated and real-world environments and eliminating key barriers in autonomous driving.

- We introduce two discriminators that learn the discrepancies between the simulation and real-world domains, enabling the transfer of knowledge from the simulation model to the real-world model.

- We propose a novel back-propagation strategy that accelerates adversarial training and improves convergence stability.

- Extensive experiments on real-world scenarios demonstrate that our method achieves state-of-the-art performance, significantly outperforming existing approaches under the new data-splitting strategy for the nuScenes dataset.

## 2 RELATED WORKS

### 2.1 END-TO-END AUTONOMOUS DRIVING MODEL TRAINED ON REAL-WORLD SCENARIO DATA

Most industry solutions utilize real-world data to train models, making them easier to apply in autonomous vehicles and minimizing the gap between training and inference. Pioneering research employs a three-layer network trained on real camera data to determine the vehicle's driving direction Pomerleau (1988). Some approaches, such as Prakash et al. (2021), directly predict planning trajectories without interacting with perception and motion prediction, which does not align with our expectations in real environments. Subsequently, Casas et al. Casas et al. (2021) achieves more accurate planning results by leveraging intermediate representations and forecasting the future states of dynamic agents. Unlike previous works, Cui et al. (2021) optimizes the diversity of the downstream ego-vehicle contingency planner to improve safety. Inspired by Qian et al. (2024), some researchers incorporate accumulated features or additional semantic occupancy supervision to enhance final outcomes Sadat et al. (2020); Hu et al. (2022), while these methods have limitations in real-world scenarios. Note that, UniAD Hu et al. (2023) introduces a unified framework that facilitates cooperation among separate tasks, including perception, prediction, and planning in the field of autonomous driving. It employs queries to decode and interact across multiple task outputs. Furthermore, Jiang et al. (2023); Ye et al. (2023); Chen et al. (2024a) predict motion planning by integrating perception results (tracking objects and map elements), and refining the planning outcomes with additional constraints, these methods demonstrate superior performance in real-world road tests. Unlike previous methods, SparseDrive Sun et al. (2024) extracts features using sparse feature sampling instead of computationally expensive BEV (bird's eye view), it also utilizes a parallel design for the motion planner, incorporating a collision-aware rescore module to enhance real-world autonomous driving. Consequently, Zhang et al. (2024) uses a unified module paired with distinct expert designs, to enhance the outcomes, offering an efficient and straightforward industry solution. Additionally, researchers have introduced Vision-Language Model (VLM)-guided methods Shao et al. (2023) in the autonomous driving context, framing the autonomous driving task as a vision-language problem Wang et al. (2024); Xu et al. (2024a).

### 2.2 END-TO-END AUTONOMOUS DRIVING MODEL TRAINED ON SIMULATED SCENARIO DATA

However, obtaining real-world data is costly, inefficient, and lacks hard cases and corner cases. To investigate the performance of end-to-end autonomous driving models in hard cases and large-scale data, researchers utilize the simulated data from virtualization engines such as CARLA Dosovitskiy et al. (2017) and LGSVL Rong et al. (2020) for training and evaluation. Codevilla et al. (2019) introduces an auxiliary speed prediction task to facilitate lateral and longitudinal direction operation. Chen et al. (2019) trains a teacher model with privileged information and utilizes this teacher model to supervise a vision-based sensorimotor agent, significantly enriching the student models with privileged information. Note that, camera-based methods suffer from inaccurate spatial information, thus, some researchers integrate multi-modal sensor data to achieve an outstanding performance Chitta et al. (2022); Liu et al. (2024). Inspired by Jiang et al. (2023); Zhang et al. (2023), VADv2 Chen et al. (2024a) employs a large planning vocabulary and simulation training data to estimate the probability distribution of planning trajectories based on the driving behavior. Similarly, PlanT Renz et al. (2022) adopts a transformer as the main module, demonstrating fast inference speed and scalability on CARLA-collected datasets. Furthermore, some methods enhance feature representations by refining the transformer module Chitta et al. (2021); Zhang et al. (2024). Consequently, ThinkTwice Jia et al. (2023b) predicts a coarse-grained future position and retrieves a salient feature from the encoder to extend the trajectory capacity of this model. The availability of simulation environments has led to significant advancements in reinforcement-based end-to-end autonomous driving methods in recent years. These methods train a teacher model with a reinforcement learning strategy to simulate diverse scenarios, such as extreme weather, non-compliant driving habits, car accidents, etc.. Guided by this teacher model, the student model can effectively learn from corner cases Zhang et al. (2021); Wu et al. (2022); Jia et al. (2023a). DriveGPT4 Xu et al. (2024b) leverages LLMs to develop a low-level vehicle control-based end-to-end autonomous driving solution. Furthermore, LLM-based autonomous driving models enable closed-loop end-to-end autonomous driving in simulators. These methods integrate localization, perception, and decision-

making into an LLM model, which outputs driving instructions Wang et al. (2023a); Shao et al. (2023); Sima et al. (2025). Additionally, Cui et al. (2024) proposes a novel benchmark to systematically evaluate various scenarios, enhancing performance and exploring the potential of LLMs.

### 2.3 Domain Adversarial Transfer

Numerous studies demonstrate the effectiveness of transfer learning by examining the transferability of deep neural networks trained on public datasets Kornblith et al. (2019). Hendrycks et al. (2019) investigates the transfer pre-training mechanism, showing that pre-training improves adversarial robustness by 10% over the base model. However, a major focus of transfer learning is unsupervised domain adaptation, which adapts a model trained on a labeled dataset to an unlabeled domain. Among these approaches, Long et al. (2013) introduces marginal and conditional distributions to construct new feature representations for unsupervised domains. MetaAlign Wei et al. (2021) proposes an effective meta-optimization strategy that maximizes the neural network gradients across two tasks during training. Yang et al. (2024) proposes an adaption transfer method to grasp novel target objects, this method leverages object attributes to facilitate robotic grasping and rapid adaptation to new domains.

Furthermore, a new strategy for style transfer has emerged. Goodfellow et al. (2014) pioneers the adversarial generative model, which simultaneously trains a generative model to capture the data distribution and a discriminative model to estimate the probability of samples. This method learns deep representations without requiring extensively annotated training data. Another approach directly controls the data generation process by incorporating additional information, such as class labels or multimodal data Mirza & Osindero (2014). Instead of learning a generic generative model, Pix2Pix Isola et al. (2018) learns a conditional generative model, which is widely used for image-to-image translation tasks, this simple framework is sufficient to fit the expected distribution using a simple loss function. Subsequent works successfully transfer style between domains using various techniques, such as stacking and additional branches Li et al. (2017); Kim et al. (2017); Luan et al. (2017). Recently, Kim et al. (2023) investigates randomized decision rules, and introduces an empirical Bayes-like method in the training process to improve the diversity of outputs. Furthermore, DTSGAN Li et al. (2024) introduces a spatio-temporal generative adversarial network that captures motion and content distribution in video sequences. Ma et al. (2025) proposes a reconstructive transfer learning approach built on a generative adversarial network (GAN) to explore the potential of the novel masked auto-encoder (MAE) image reconstruction model. This approach achieves a significant improvement compared to other GAN-based methods. Additionally, some researchers incorporate GANs into reinforcement learning tasks to enhance robustness. Among them, Xie et al. (2025) proposes a dual-agent adversarial policy learning model, enabling agents to spontaneously learn semantics without additional human prior knowledge. Experiments demonstrate that this adversarial framework significantly improves performance.

## 3 Problem Statement

### 3.1 Limitations of Real-World Data

In real-world autonomous driving application scenarios, researchers collect data from vehicles, these vehicles are driven by skilled people. This leads to three key challenges: (1) Autonomous driving is hindered by data collection and data labeling in the physical world, both of which require substantial human and financial resources; (2) Rare scenarios, such as traffic accidents, driving violations, non-standard roads, pedestrian trespassing, and extreme weather conditions, are challenging to capture. (3) There is a lack of high-order interactions and efficient decision-making between vehicles in real-world environments. As imitation learning-based approaches, current autonomous driving methods cannot guarantee vehicle safety and reliability across all driving scenarios unless these issues are addressed.

### 3.2 Limitations of Simulation Data

Recently, simulations have become widely used in autonomous driving research. Autonomous driving simulators support flexible configurations of sensor suites, kinetic models, interactive static and

dynamic scenes, environmental conditions (such as weather and lighting), and even detailed collision data. These simulations effectively address some of the real-world challenges mentioned above. Simulations facilitate the collection of large-scale labeled data, which can be utilized to validate the effectiveness of data-driven models. Furthermore, since certain rare and challenging scenarios are too dangerous to stage in real-world settings, simulators can generate them freely. Additionally, autonomous driving models can undergo closed-loop evaluations in simulators, making them more representative of real-world scenarios. Researchers are increasingly focused on developing more realistic simulation environments. However, a significant gap remains between simulations and real-world driving. In summary, simulations introduce a new challenge: how to transfer models trained on virtual datasets to real-world applications.

### 3.3 METHODS SHIFT

Autonomous driving methods face the domain shift problem, where models perform well in simulated environments during training but struggle in real-world driving scenarios. To address these challenges, we propose a novel method that simulates non-standard driving scenarios and integrates them into our model to enhance reliability. We train our model using a novel adversarial strategy to transfer simulated scenarios into real-world applications.

## 4 METHOD

### 4.1 DISCREPANCY BETWEEN SIMULATION AND REAL-WORLD

We assume that the data space samples from the simulation and real-world domains are represented as $D_S = \{(x_i^s, y_i^s)\}$ and $D_R = \{(x_i^r, y_i^r)\}$ respectively. Each domain corresponds to a joint distribution over the input space $X$ and output space $Y$. We define the hypothesis parameter sets $H_s$ for the simulation domain, which is used to predict simulation data labels.

$$H_s := \{h_s : D_S\} \tag{1}$$

The intuition behind the above definitions and assumptions is that we use the mapping $T_{s2r}$ to minimize the distance between simulation and real-world domains. The hypothesis $h$ mentioned above is employed to predict the result $\hat{y}$, and we define the transformation function as follows:

$$h_r = T_{s2r} \cdot h_s \tag{2}$$

Based on this formulation, we define the hypothesis set for the real-world domain as:

$$H_r := \{h_r : D_R | h_r(\cdot) = h_s(T_{s2r}(\cdot)), h_s \in H_s\} \tag{3}$$

where the real-world hypothesis parameter sets $H_r$ is represented in terms of $h_s$ and transfer matrix $T_{s2r}$.

In our method, the first process is to quantify the discrepancy between both domains. Given the space samples of the two domains, the discrepancy $D^\delta$ between two domains $D_S$ and $D_R$ is defined as follows:

$$D_h^\delta(D_S||D_R) := \mathop{\mathbb{E}}_{x^s \in D_S} l_s(h_s(x^s), y^s) - \mathop{\mathbb{E}}_{x^r \in D_R} l_r(h_r(x^r), y^r) \tag{4}$$

This indicates that the hypothesis function $h$, which is used to fit the distribution of both data, directly influences the discrepancy. $l_s$ and $l_s$ are used to compute the losses. As described in these equations, the upper bound in terms of the simulation domain can be presented by the following.

$$\sup_{h_s \in H_s} \mathop{\mathbb{E}}_{x^s \in D_S} [l_s(h_s(x^s), y^s)] := \rho < \infty \tag{5}$$

Here, $l_s$ is the bounded continuous loss, $\rho$ is the expected parameter. Since we have eq. (2), we further explore the formula for discrepancy $D^\delta$:

$$D_h^\delta(D_S||D_R) = \sup_{h_s \in H_s} ( \mathop{\mathbb{E}}_{x^s \in D_S} (h_s(x^s) - y^s) -$$
$$\mathop{\mathbb{E}}_{x^r \in D_R} (\hat{T}_{s2r}(h_r(x^r) - y^r))) \tag{6}$$

where $sup$ denotes supremum, $\hat{T}_{s2r}$ is the Fenchel conjugate of a lower semi-continuous convex function. Notably, this discrepancy $D_h^\delta$ is a variational formulation of the f-divergence for the convex function $\delta$, thus, $D_h^\delta(D_S||D_R)$ serves as a lower bound estimate of the f-divergence function.

In our method, the hypothesis is divided into two components: the perception hypothesis and the motion hypothesis. Consequently, $D_h^\delta(D_S||D_R)$ is decomposed into $D_p^\delta(D_S||D_R)$ and $D_m^\delta(D_S||D_R)$, corresponding to perception and motion discrepancies, respectively.

## 4.2 ADVERSARIAL OPTIMIZATION

To transfer the simulation domain to the real-world domain, we optimize three components of our model: the end-to-end multi-task network, the perception discriminator $D_p^\delta(D_S||D_R)$ and the motion discriminator $D_m^\delta(D_S||D_R)$. These two discriminators estimate the discrepancy between $D_S$ and $D_R$ aiming to distinguish between the features of two domains. We summarize the total optimization problem as follows:

$$\min_{\delta} \mathop{\mathbb{E}}_{x \in D_S, D_R} [l(h(x), y)] + D_p^\delta(D_S||D_R) + D_m^\delta(D_S||D_R) \tag{7}$$

where, $l$ is the multi-task combination loss, $h$ is the multi-task optimization function. Furthermore, we define an upper bound for these two discriminators:

$$d_p^\delta = \mathop{\mathbb{E}}_{x \in D_S} [log(D_p^\delta(f_p(x)))] + \mathop{\mathbb{E}}_{x \in D_R} [log(1 - D_p^\delta(f_p(x)))] \tag{8}$$

$$d_m^\delta = \mathop{\mathbb{E}}_{x \in D_S} [log(D_m^\delta(f_m(x)))] + \mathop{\mathbb{E}}_{x \in D_R} [log(1 - D_m^\delta(f_m(x)))] \tag{9}$$

where $f_p$ and $f_m$ are the multi-task networks responsible for extracting features from the input data. From eq. (8) and eq. (9), we derive:

$$D_p^\delta(D_S||D_R) + D_m^\delta(D_S||D_R) <= \max_{\delta} d_p^\delta + \max_{\delta} d_m^\delta \tag{10}$$

The final optimized function is given by:

$$\min_{\delta} \max_{\delta} \mathop{\mathbb{E}}_{x \in D_S, D_R} [l_r(h_r(x), y)] + d_p^\delta + d_m^\delta \tag{11}$$

As shown in eq. (11), our model first minimizes the multi-task loss functions using finite samples, the second and third components correspond to discriminator losses, which are optimized using an adversarial strategy with a min-max formulation.

## 4.3 MODEL ARCHITECTURE

As depicted in fig. 1, our simulation to real-world transfer learning model consists of three parts: the simulation model, the real-world model and discriminators.

**Simulation Model.** The simulation model is a complete end-to-end autonomous driving framework trained with simulation datasets. Its primary objective is to learn various hard cases, such as an accident ahead, vehicles driving violation, etc. This model primarily follows the SparseDrive Sun et al. (2024). Firstly, it extracts features from 6 surrounding cameras using ResNet He et al. (2015), followed by a transformer decoder to extract perception features, including dynamic and static object representations. Using these feature representations, a unified motion decoder, cascaded with multiple transformer layers, predicts both ego-motion planning trajectories and agent prediction trajectories simultaneously.

**Real-world Model.** We also incorporate a real-world model, structurally identical to the simulation model but initially trained on real-world datasets along with a subset of simulation data. This model is designed to transfer learned simulation scenarios to real-world applications and serves as the final inference model.

**Discriminators.** Inspired by GAN-based methods, our discriminators estimate the distributional discrepancy between the two data domains and adapt the simulation model's performance to align

with real-world expectations. These discriminators consist of two modules: the perception discriminator and the motion discriminator, both are implemented using transformers. The perception discriminator processes instance features from both the simulation and real-world models. Its role is to minimize the domain gap and adversarially transfer the expected perception representations to the real-world model. Similarly, the motion discriminator processes motion features from both models to facilitate adversarial transfer of expected motion trajectory representations to the real-world model.

### 4.4 LOSS

Different from existing methods, our approach employs distinct loss functions at different training stages.

**Pre-training Loss.** Firstly, during the pre-training step, we adopt a multi-task loss similar to Sun et al. (2024) for both the simulation model and real-world model, which serves to pre-train our base model. The multi-task loss is defined as follows:

$$l_{pre} = l_{det} + l_{map} + l_{mot} + l_{pred} + l_{plan} \tag{12}$$

where $l_{det}$ and $l_{map}$ denotes the detection loss and online map construction loss, $l_{mot}$ is the tracking loss, $l_{pred}$ and $l_{plan}$ correspond to the prediction loss and planning loss, respectively.

**Adversarial Training Loss.** Secondly, in step 1 of adversarial training, we incorporate two discriminator losses: the perception discriminator loss and the motion discriminator loss. These losses enable the discriminators to learn the differences between the two data distributions, and they are defined as follows:

$$l_{step1} = l_{pd} + l_{md} \tag{13}$$

Here, $l_{pd}$ and $l_{md}$ are perception discriminator loss and motion discriminator loss, respectively. In step 2 of adversarial training, we use both the discriminator loss and the real-world multi-task loss, as shown below:

$$l_{step2} = l_{det} + l_{map} + l_{mot} + l_{pred} + l_{plan} + l_{pd} + l_{md} \tag{14}$$

However, as described in the Model Training section, two discriminators only pass the gradient parameters, and do not update their model parameters.

### 4.5 MODEL TRAINING

Our training process consists of two stages: pre-training and adversarial training.

#### 4.5.1 PRE-TRAINING

In this training process, we train our simulation model with simulation datasets. Since the simulation datasets contain a large number of hard cases, such as accidents ahead, vehicle driving violations, non-standard roads, and pedestrian trespassing, it is designed to handle complex scenarios effectively. Next, we train our real-world model using both simulation and real-world datasets, to establish a foundation model for rapid convergence during adversarial training.

#### 4.5.2 ADVERSARIAL TRAINING WITH MULTI-STEP BACK-PROPAGATION

The adversarial training process is a crucial component of our model, as detailed in fig. 1. We load the pre-trained model for our simulation and real-world model firstly. The adversarial training process involves multiple steps with our unique back-propagation strategies.

**Step 1**

Forward Propagation: In this forward propagation, the simulation and real-world models are frozen, and only the discriminators are trained. We feed simulation data and real-world data into simulation model and real-world model separately, then the features extracted from both models are subsequently fed into the discriminators.

Back Propagation: When our discriminators outputs the final logits, we used discriminator losses to perform the backward propagation. Since only the discriminators are trainable at this stage, we pack-propagate and update exclusively the discriminator parameters, while the simulation and real-world models remain unchanged.

**Step 2**

Forward Propagation: In this process, only the real-world model and discriminators are involved. Both real-world and simulation data are fed into the real-world model, and the resulting features are passed through the discriminators to obtain logits.

Back Propagation: The backward propagation in this step is our unique contribution. The backward propagation process is different from others. Firstly, we only back-propagate the discriminator losses pass through the two discriminators separately without updating parameters of two discriminators. Secondly, the back-propagated gradients from discriminators and the multi-task supervised losses are used to back-propagate and update the real-world model. This strategy mitigates convergence instability and accelerates adversarial training.

Table 1: **Open-loop planning performance on the new split nuScenes dataset.** AT-Drive achieves a significant performance improvement over other methods across evaluation metrics. The best results achieved by our approach are highlighted in bold. *: Reproduced with simulation dataset pre-trained, real-world data finetuned. These experiments are reproduced on the new split nuScenes and simulation dataset, with all evaluations performed using NVIDIA Tesla A800 GPUs.

| Methods | L2 (m) ↓ | | | | Collision Rate (%) ↓ | | | |
|---|---|---|---|---|---|---|---|---|
| | 1s | 2s | 3s | Avg. | 1s | 2s | 3s | Avg. |
| UniAD Hu et al. (2023) | 0.57 | 0.86 | 1.40 | 0.94 | 0.67 | 0.75 | 1.03 | 0.82 |
| VAD Jiang et al. (2023) | 0.55 | 0.78 | 1.23 | 0.85 | 0.26 | 0.33 | 0.54 | 0.38 |
| VADv2 Chen et al. (2024b) | 0.55 | 0.75 | 1.21 | 0.84 | 0.24 | 0.34 | 0.50 | 0.36 |
| SparseDrive Sun et al. (2024) | 0.56 | 0.75 | 1.19 | 0.83 | 0.18 | 0.31 | 0.47 | 0.32 |
| SparseDrive-finetune* | 0.59 | 0.71 | 1.17 | 0.82 | 0.18 | 0.30 | 0.50 | 0.33 |
| Ours-finetune* | 0.45 | 0.72 | 1.07 | 0.75 | 0.13 | 0.21 | 0.46 | 0.27 |
| Ours | **0.33** | **0.64** | **0.90** | **0.62** | **0.06** | **0.11** | **0.19** | **0.126** |

## 5 EXPERIMENTS

### 5.1 DATASETS

**Real-world Datasets.** We use the nuScenes Caesar et al. (2020) as our real-world datasets, it consists of 1000 scenes, annotated at 2 Hz. These datasets were collected with multi-modal sensors and labeled per key frame.

However, the official training/validation split has an overlap of over 84% of scenes between the training and validation sets, which means that the hard cases are equally divided in the training and validation datasets. In our paper, we aim to train our model to enhance robustness in unfamiliar scenarios (hard cases); therefore, we adopt a new training-validation split for the nuScenes dataset. We designate relatively simple scenes, such as straight roads, visible lane markings, and multi-lane highways, as the training dataset, while challenging scenarios, such as turns, intersections, unmarked roads, and narrow passages, constitute the validation dataset. This split strategy enables us to evaluate whether our method can effectively transfer hard cases from the simulation environment to real-world scenarios.

**Simulation Datasets.** This dataset is derived from the CARLA Leaderboard v2 simulator Dosovitskiy et al. (2017) and is referred to as the CARLA dataset. Since we use the same model as the real-world model, we configured the sensors similarly to those in nuScenes (one GNSS, one LiDAR, six surrounding cameras, and HD maps) and segmented the dataset into multiple clips, following the structure of nuScenes. Specifically, following Jia et al. (2024), we simulate a large number of hard cases, including accidents ahead, vehicle traffic violations, non-standard roads, and pedestrian trespassing. These hard cases are intended to be transferred to the real-world model.

### 5.2 IMPLEMENTATION DETAILS

We use Sun et al. (2024) as our baseline. The training process is complex and challenging in terms of convergence; therefore, we utilize eight A800 GPUs with a large total batch size to prevent the model from getting trapped in local optima or experiencing complete divergence. Additionally, we employ different optimizers for the baseline model and discriminators to enhance performance.

## 5.3 COMPARISONS WITH STATE-OF-THE-ART METHODS

Since our goal is to explore transfer learning from simulation to real-world scenarios and address the scarcity of rare and challenging scenarios in real-world datasets, we employ a new split strategy for the nuScenes dataset. This new split strategy assigns challenging scenarios (hard cases) to the evaluation dataset to assess the ability of our method to transfer hard cases effectively.

To ensure a fair comparison with other state-of-the-art (SOTA) methods, we retrained these models using our new split strategy on the nuScenes dataset. As shown in table 1, we compare the performance of our approach with existing methods. All methods are evaluated on the newly split nuScenes dataset. Our approach demonstrates superior performance on the newly split dataset. Specifically, AT-Drive surpasses competing methods by more than 25.3% and 60.6% in terms of average L2 distance error and average collision rate, respectively. These results strongly validate the effectiveness of our novel model design.

Table 2: Ablation studies were conducted on the key design elements of AT-Drive using the new split real-world dataset and simulation dataset. The results prove the effectiveness of our design.

| Index | Module Components | L2(m) ↓ | Collision Rate(%) ↓ |
|---|---|---|---|
| 1 | Simulation baseline: 
 Simulation dataset trained, real-world dataset tested | 11.32 | 27.81 |
| 2 | Real-world baseline: 
 Real-world dataset trained, real-world dataset tested | 0.80 | 0.31 |
| 3 | Simulation dataset pre-trained 
 real-world dataset finetuned | 0.75 | 0.27 |
| 4 | Based on two baselines + Motion discriminator | 0.73 | 0.21 |
| 5 | Based on two baselines + Perception discriminator | 0.67 | 0.22 |
| 6 | Ours (Based on two baseline + Two discriminators) | 0.62 | 0.126 |

## 5.4 ABLATION STUDY

### 5.4.1 KEY COMPONENTS DESIGN

We validate our components by performing an ablation study to prove the effectiveness of our model design. As shown in table 2, we start by validating our baseline models: the simulation baseline (only the simulation model) and the real-world baseline (only the real-world model). Specifically, in Index-1, we train our simulation baseline model on the simulation dataset and evaluate it on the real-world validation dataset. Notably, the baseline (Index-1) demonstrates extremely poor performance. In the second row (Index-2), we train our real-world model on the real-world dataset. However, since hard cases were assigned to the validation dataset and easy cases to the training dataset, the model's performance is suboptimal, achieving only 0.80 in L2 distance and 0.31 in collision rate. We also pre-trained our baseline with a simulation dataset, and finetuned with the real-world dataset, achieving 0.73 in L2 distance and 0.27 in collision rate. As shown in Index-4, using only the motion discriminator yields an L2 distance of 0.73 and a collision rate of 0.21. Incorporating the perception discriminator further enhances feature representation, reducing the L2 distance by 0.11 and the collision rate by 0.1.

## 6 CONCLUSION

In this study, we proposed an end-to-end autonomous driving method that transfers simulated imitation driving capabilities to real-world deployment. The approach comprises two base models and two discriminators. The simulation model learns uncommon scenarios from synthetic data and transfers this knowledge to the real-world model through adversarial learning with discriminators. Importantly, we designed an efficient adversarial training process, which is critical for the model's successful optimization. Our method effectively addresses hard cases in real-world scenarios. We evaluated our method on the newly split nuScenes dataset, and the experimental results demonstrate that AT-Drive significantly outperforms existing methods in complex driving scenarios.

## 7 REPRODUCIBILITY STATEMENT

our method is reproducible, we provide comprehensive instructions and code, reproducing large-scale experiments. All datasets used are publicly available, and we provide preprocessing scripts

where necessary. Hyperparameters, training details, and evaluation protocols are described in the paper and included in the code repository. Our experiments can be reproduced on a 8 GPU within the reported computational budget.

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

# A  APPENDIX

## A.1  PERCEPTION AND PREDICTION PERFORMANCE

To further evaluate our method, we conducted several sub-task experiments on the newly split nuScenes validation set. Below, we compare our results with state-of-the-art methods.

**Detection.** We conduct a 3D detection evaluation experiment. As shown in table 3, AT-Drive achieves 0.437 mAP, 0.572 mATE, 0.258 mASE, 0.402 mAOE, 0.255 mAVE, 0.170 mAAE and 0.563 NDS, surpassing SparseDrive Sun et al. (2024) and UniAD Hu et al. (2023) over all metrics. These results demonstrate the effectiveness of our model design.

**Multi-object Tracking.** Since our method is an end-to-end autonomous driving approach, we generate tracking results to enhance the continuity of temporal and spatial features. The evaluation metrics for tracking are detailed in table 4. Our method achieves 0.467 AMOTA, 1.143 AMOTP, 0.579 Recall, and 656 IDS, outperforming existing methods by 29.0%, 14.1%, 25.1%, and 8.8%, respectively.

**Online mapping.** The results of the online mapping are illustrated in table 5, AT-Drive delivers the expected performance, achieving a 22.3% improvement in Mean Average Precision ($mAP$) compared to SparseDrive Sun et al. (2024). Specifically, AT-Drive significant increase in all map elements AP, yielding 40.1, 43.3 and 47.0 in terms of pedestrian crossings, lane dividers, and road boundaries.

**Prediction.** Agent trajectory prediction significantly influences the final ego motion planning. Therefore, we evaluate the prediction results to assess our method. table 6 illustrates that our model outperforms the existing method by 0.09, 0.28, 0.008 and 0.066 in minADE, minFDE, MR and EPA respectively.

Table 3: **Object detection on the new split nuScenes dataset.** AT-Drive achieves the best performance on detection tasks. All other methods are re-implemented on the new split nuScenes dataset for a fair comparison.

| Methods | mAP ↑ | mATE ↓ | mASE ↓ | mAOE ↓ | mAVE ↓ | mAAE ↓ | NDS ↑ |
|---|---|---|---|---|---|---|---|
| UniAD Hu et al. (2023) | 0.311 | 0.606 | 0.277 | 0.638 | 0.306 | 0.184 | 0.455 |
| SparseDrive Sun et al. (2024) | 0.396 | 0.581 | 0.282 | 0.505 | 0.284 | 0.175 | 0.515 |
| Ours | **0.437** | **0.572** | **0.258** | **0.402** | **0.255** | **0.170** | **0.563** |

Table 4: **Multi-object tracking.** AT-Drive achieves competitive performance against state-of-the-art methods, Experiments are reproduced on the new split nuScenes.

| Methods | AMOTA ↑ | AMOTP ↓ | Recall ↑ | IDS ↓ |
|---|---|---|---|---|
| UniADHu et al. (2023) | 0.300 | 1.454 | 0.387 | 939 |
| SparseDriveSun et al. (2024) | 0.362 | 1.331 | 0.463 | 719 |
| Ours | **0.467** | **1.143** | **0.579** | **656** |

Table 5: **Online mapping on the new split nuScenes dataset.** Comparison with state-of-the-art method, AT-Drive outperforms other methods over all metrics.

| Methods | $AP_{ped}$ ↑ | $AP_{div}$ ↑ | $AP_{bound}$ ↑ | mAP ↑ |
|---|---|---|---|---|
| VADJiang et al. (2023) | 27.5 | 24.6 | 40.8 | 31.0 |
| SparseDriveSun et al. (2024) | 32.7 | 35.2 | 39.8 | 35.9 |
| Ours | **40.1** | **43.3** | **47.0** | **43.9** |

## A.2 CLOSED-LOOP EVALUATION

Since our goal is to transfer the model's ability to handle challenging scenarios into real-world settings, it is essential to perform closed-loop evaluations in real-world environments. To this end, we adopted the open-source NAVSIM [6].

For a fair comparison with existing methods on the NAVSIM dataset, we used the standard navtrain and navtest splits for training and validation, respectively, as these splits were also used in prior work. We followed the community's recommended PDMS metric for final evaluation. As shown in table 7, our method achieves notable improvements over the existing ParaDrive baseline.

Furthermore, due to limitations in experimental facilities and road safety regulations, we were unable to conduct certain experiments, such as those involving perturbations or sensor noise robustness.

## A.3 EVALUATION ON ORIGINAL SPLIT NUSCENES DATASET

We provide a comparison of AT-Drive's performance against existing methods on the original split nuScenes dataset. As shown in table 8, our method achieves an average L2 distance of 0.61 and an average collision rate of 0.09, which are slightly better than the competing methods by 0.02 and 0.01, respectively. When combined with the main experiments conducted above, the results indicate that while our method excels in identifying and handling hard cases, it shows limited improvement in normal scenarios, which have already been extensively studied during the training process. Swerdlow et al. (2024); Wen et al. (2024); Gao et al. (2024; 2025); Wang et al. (2023b); Li et al. (2023)

## A.4 EVALUATION ON STRAIGHT REGRESSOR BASELINE

we design a new dummy regressor baseline based on our method, termed Straight Regressor Baseline. To produce a straight trajectory, we ignore the y-axis, only predict the x-axis trajectory values, whereas existing methods predict both (x, y) coordinates. We trained and evaluated this Straight Regressor Baseline on new nuScenes split, the results are illustrated in table 9.

Table 6: **Prediction.** AT-Drive achieves competitive performance against state-of-the-art perception-oriented methods on the new split nuScenes dataset.

| Methods | minADE(m)↓ | minFDE(m)↓ | MR↓ | EPA↑ |
|---|---|---|---|---|
| UniADHu et al. (2023) | 0.78 | 1.17 | 0.142 | 0.462 |
| SparseDriveSun et al. (2024) | 0.76 | 1.35 | 0.145 | 0.445 |
| Ours | **0.67** | **1.07** | **0.137** | **0.511** |

Table 7: **Experiments on NAVSIM.**

| Methods | PMDS |
|---------|------|
| UniAD | 83.4 |
| ParaDrive | 84.0 |
| Ours | 85.3 |

Table 8: **Open-loop planning performance on the original split nuScenes dataset.** AT-Drive achieves a slight improvement over other methods across evaluation metrics. These experiments are reproduced on the original split nuScenes and simulation dataset. *: Official checkpoint re-validated with corrected metrics.

| Methods | L2 (m) ↓ | | | | Collision Rate (%) ↓ | | | |
|---------|------|------|------|------|------|------|------|------|
| | 1s | 2s | 3s | Avg. | 1s | 2s | 3s | Avg. |
| ST-P3Hu et al. (2022) | 1.33 | 2.11 | 2.90 | 2.11 | 0.23 | 0.62 | 1.27 | 0.71 |
| UniADHu et al. (2023)* | 0.45 | 0.70 | 1.04 | 0.73 | 0.62 | 0.58 | 0.63 | 0.61 |
| VADJiang et al. (2023)* | 0.41 | 0.70 | 1.05 | 0.72 | 0.07 | 0.17 | 0.41 | 0.22 |
| PPADChen et al. (2024c) | 0.30 | 0.69 | 1.26 | 0.75 | 0.03 | 0.22 | 0.73 | 0.33 |
| ParaDrive Weng et al. (2024) | 0.26 | 0.59 | 1.12 | 0.66 | 0.00 | 0.12 | 0.65 | 0.26 |
| SparseDriveSun et al. (2024) | 0.29 | 0.63 | 0.97 | 0.63 | 0.03 | 0.09 | 0.19 | 0.10 |
| Ours | 0.24 | **0.60** | **0.84** | **0.56** | 0.03 | **0.07** | **0.13** | **0.076** |

This result demonstrates that the new split of the nuScenes dataset, which includes challenging scenarios, effectively shows that our strategy successfully transfers challenge handling capabilities to the real-world model.

Table 9: **Performance of Straight Regressor Baseline on new nuScenes split.**

| Methods | L2 (m) ↓ | | | | Collision Rate (%) ↓ | | | |
|---------|------|------|------|------|------|------|------|------|
| | 1s | 2s | 3s | Avg. | 1s | 2s | 3s | Avg. |
| Straight Regressor Baseline | 0.82 | 1.10 | 2.07 | 1.33 | 0.74 | 1.18 | 1.65 | 1.19 |
| Ours | **0.33** | **0.64** | **0.90** | **0.62** | **0.06** | **0.11** | **0.19** | **0.126** |

We also evaluated the Straight Regressor Baseline on the original nuScenes split. The results in table 10, indicate that the new split strategy is significantly more challenging than the original nuScenes split.

## A.5 GRANULAR STUDIES

In our main method, the ratio of simulation data vs real-world data is 0.6 (600:1000), we added an experiment of 0.3. for ratio 0.3, we have achieved the results in table 11.

## A.6 QUALITATIVE ANALYSIS

We conducted additional qualitative comparisons to systematically evaluate the performance of our AT-Drive method. fig. 2 and fig. 3 illustrate a continuous bypass scenario. In fig. 2, our method bypasses the vehicles on the left and prepares to return to its lane. However, when another vehicle appears ahead in the path, as shown in fig. 3, the ego vehicle aborts the lane return and executes a new bypass maneuver. fig. 4 depicts a complex intersection scenario, where crosswalk markings could mislead trajectory planning. However, our method successfully generates an optimal path, as observed in fig. 4. fig. 5 illustrates a turning scenario surrounded by vehicles. The qualitative results indicate that our method generates an effective trajectory plan.

AT-Drive's ability to transfer rare and challenging scenarios knowledge enables it to handle complex trajectory predictions. As shown in fig. 6, at a road junction, the vehicles behind (captured in the rear and rear-right camera views) are positioned at a critical point for lane switching. Our method generates complex predicted trajectories, which is one of the major limitations.

Table 10: **Performance of Straight Regressor Baseline on origin nuScenes split.**

| Methods | L2 (m) ↓ | | | | Collision Rate (%) ↓ | | | |
|---|---|---|---|---|---|---|---|---|
| | 1s | 2s | 3s | Avg. | 1s | 2s | 3s | Avg. |
| Straight Regressor Baseline | 0.58 | 0.77 | 1.31 | 0.89 | 0.66 | 0.69 | 0.75 | 0.70 |
| Ours | 0.24 | **0.60** | **0.84** | **0.56** | 0.03 | **0.07** | **0.13** | **0.076** |

Table 11: **Granular Studies over simulation data vs real-world.**

| Methods | L2 | Collision Rate |
|---|---|---|
| Ours (with dataset ratio = 0.3) | 0.300 | 1.454 |

## A.7 DATASET ANALYSIS

Simulation Dataset

We have included a statistical distribution of the challenging simulation scenarios, as shown in table 12. A new supplementary table below provides detailed counts for each scenario category. We hope these additions enhance the transparency of our dataset and support a more rigorous interpretation of the model's performance in handling rare but critical driving events.

Table 12: CARLA Scenarios Statistical Distribution.

| Scenarios Category | Proportion(%) | Clips Count(total 600) |
|---|---|---|
| Normal (Straight, Left, Right Turn) | 61.6 | 370 |
| Pedestrian Intrusion | 4.7 | 28 |
| Fog | 6.7 | 40 |
| Rain | 7.8 | 47 |
| NearMiss Vehicle Interactions | 2.7 | 16 |
| Traffic Jam | 4.5 | 27 |
| Traffic Accident | 1.8 | 11 |
| Vehicle Cut In | 2.2 | 13 |
| Opposite Vehicle Intrusion | 0.8 | 5 |
| Vehicle U-Turning | 0.5 | 3 |
| Construction Obstacle | 4.0 | 24 |
| Bicycle Intrusion | 0.5 | 3 |
| Lane Merge | 1.2 | 7 |
| No Traffic Light Intersection | 0.7 | 4 |
| Turn Left and Merge In | 0.3 | 2 |

Real-World Dataset

To demonstrate our model's ability to handle uncommon cases, we used the challenging subset of the nuScenes dataset as the val set (a total of 122 scenes), while the remaining easier cases were used for training. The test set includes scenarios such as: Scene-0026_379: The ego vehicle is intercepted by a construction worker yielding to a construction truck approaching from the left. Scenes 0046_568, 0094_948, 0131_1153, and 0162_1556: Instances of pedestrian intrusion. Scene-0150_1358: A construction worker blocks the road using traffic cones or water barriers in front of the ego vehicle. Scene-0201_1978: The ego vehicle is blocked by a car attempting to park in a designated area. As shown in the last two rows of Table 1, our proposed method outperforms existing standard methods on these challenging scenarios.

## A.8 DATASET SPLITTING

Most end-to-end autonomous driving methods evaluate their performance using the nuScenes dataset, which follows the original 700/150/150 split for training, validation, and testing scenes. However, this original split was designed for normal case scenarios, not hard cases. In the original split, the training and validation sets overlap by over 84% in terms of static environments, with both easy and hard scenarios proportionally distributed across both sets. This overlap means that a model could easily memorize the static environments from the training set and perform exceptionally well on the validation set, which can lead to complete failure when generalizing to new, unseen scenes. To better assess the model's ability to generalize to unfamiliar environments and adapt to challenging cases, we propose a new dataset split, where hard cases are used for validation and easy cases for training. This approach focuses on testing the model's ability to transfer simulated hard cases to unseen real-world scenarios.

To provide a clearer understanding, we visualize both the original and the new dataset splits. As shown in fig. 7 and fig. 8, the green line represents the training set, the blue line indicates the validation set, and the red line denotes the test set. We ensure that there is no overlap between the training and validation datasets across all six cities, maintaining a balanced distribution. As depicted in these figures, our new split strategy assigns the hard cases to the validation set, thereby preventing the model from memorizing scenarios during the training process.

## A.9 METRIC ACRONYMS

We have add a subsection below to explain the metric acronyms. The short summary is below: L2(m): L2 distance, unit is meter. Avg.: Average of 1 second, 2 second and 3 second. mAP: Mean Average Precision. mATE: Mean Average Translation Error. mASE: Mean Average Scale Error. mAOE: Mean Average Orientation Error. mAVE: Mean Average Velocity Error. mAAE: Mean Average Attribute Error. NDS: nuScenes Detection Score. AMOTA: Average Multi Object Tracking Accuracy. AMOTP: Average Multi Object Tracking Precision. APped,APdiv,APbound: Average Precision of pedestrian crossings, lane dividers, and road boundaries. minADE: Minimum Average Displacement Error. minFDE: Minimum Final Displacement Error. MR: Miss Rate.

## A.10 TRAINING DETAILS

We have put the hyper-parameters in the end of our appendix, the main parameters are:

total_batch_size =64

step 1:

optimizer:AdamW,

lr=4.5e-4,

learning rate policy: CosineAnnealing,

warmup="linear",

warmup_iters=1100,

warmup_ratio=1.0 / 3,

min_lr_ratio=2.7e-4,

optimizer must use grad_clip: max_norm=42

Must use DistributedGroupSampler

step 2:

optimizer:AdamW,

lr=2.0e-5,

optimizer must use grad_clip: max_norm=23

We were tortured by the adversarial training for a period time. We were conducted experiments by adjust the learning rate, decay rate, increase batch size. We found that if the batch size is small, the two discriminator loss has large fluctuations. In the adversarial training step 1, we increase batch size to 64, then, we found our model convergence rapidly in the preliminary 3000 iters, then the loss is hard to decrease, the variance of each losses is increase, even we adjust the learning rate, it does not solve the problem, just postpone fluctuations. So, we added a gradient clipping to limit the gradient values. We choose L2 norm and a set of max norm value 42, we found the we can safely pass the fluctuations, then, after 5000 iters, the loss is declined as we expected. During the adversarial training step 2, we set gradient clipping with L2 norm and max norm value 23, then, we achieved the expected results. I think gradients conflict happened in all multi-task models, not only in our adversarial training, if we adjust hyper-parameters let the model pass the fluctuation period, the convergence will converge normally.

### A.11 LIMITATIONS

While our model significantly improves upon existing methods, it still has certain limitations. The complexity of our training process makes optimization challenging, particularly during the initial epochs. If minor deviations occur in the initial epochs, the entire training process may diverge. Moreover, maintaining two baseline models results in high training costs.

### A.12 THE USE OF LLMS

This project only uses a large language model (LLM) to correct grammatical, morphological, and syntactic errors in Polish text.

### A.13 ETHICS STATEMENT

We followed the ICLR Code of Ethics. Our work uses public datasets with proper attribution. The proposed approach does not involve experiments on human subjects, personal data collection, or sensitive attributes.

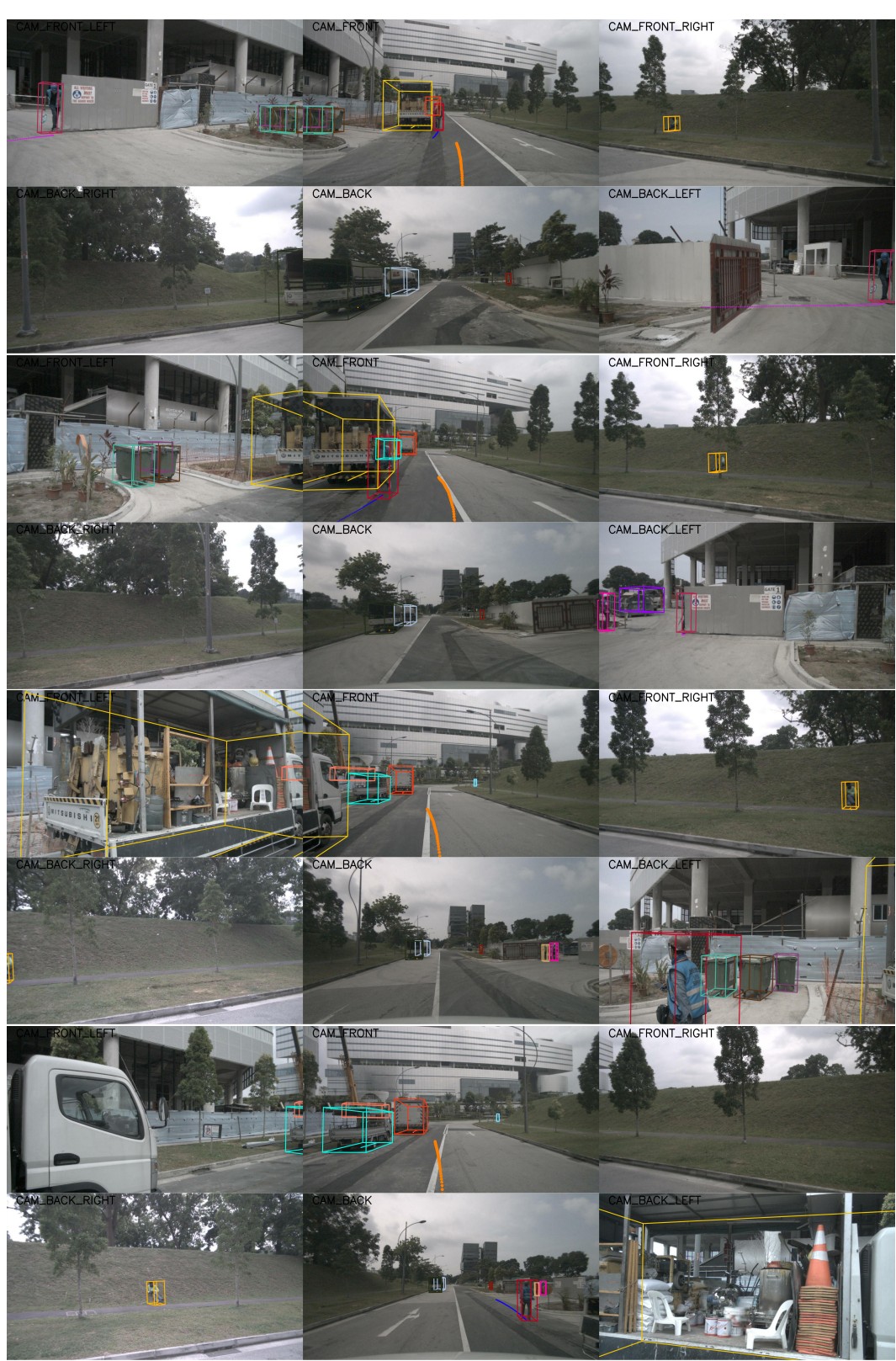

Figure 2: Qualitative results on continuity bypass case. Part 1.

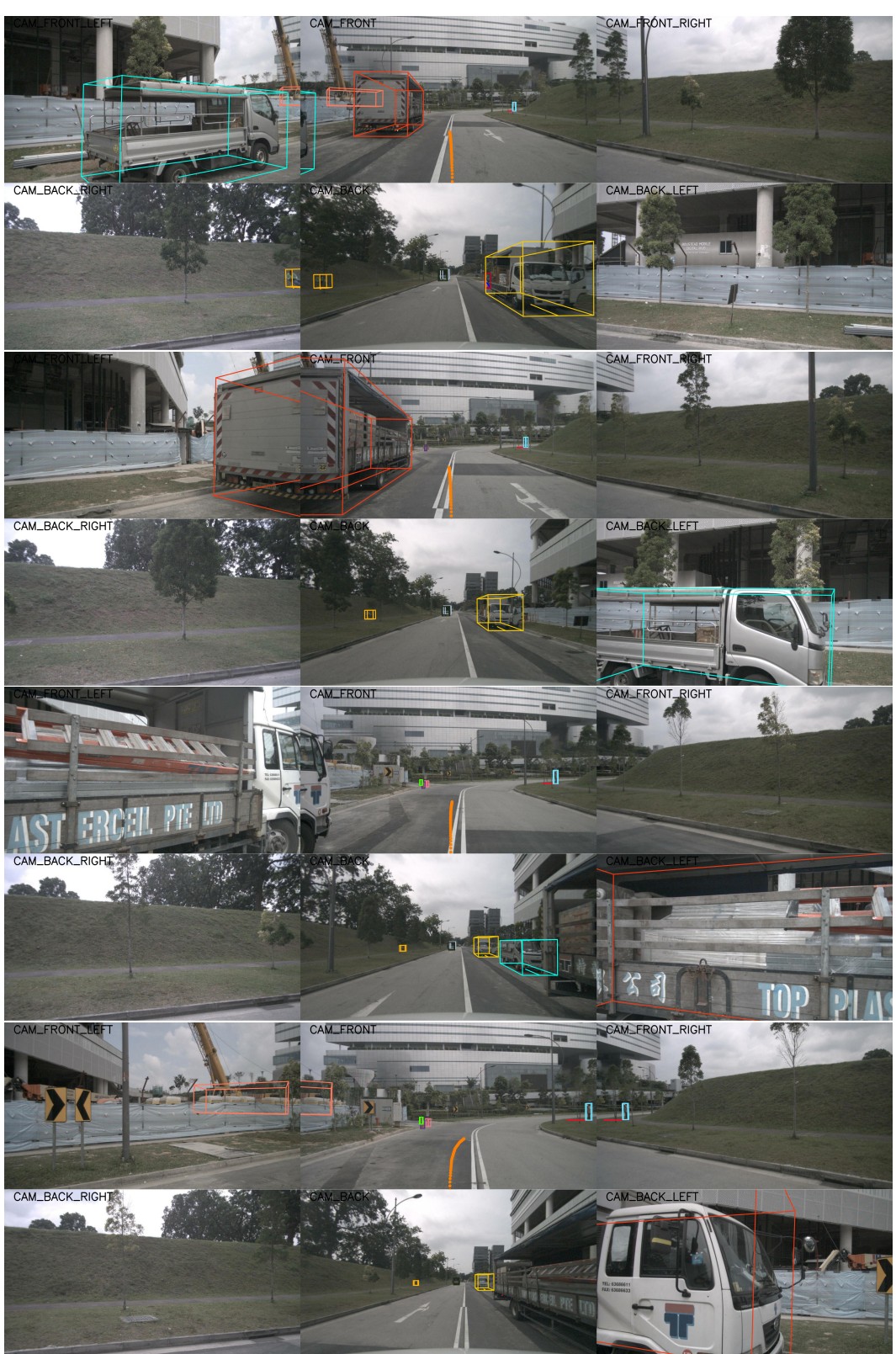

Figure 3: Qualitative results on continuity bypass case. Part 2.

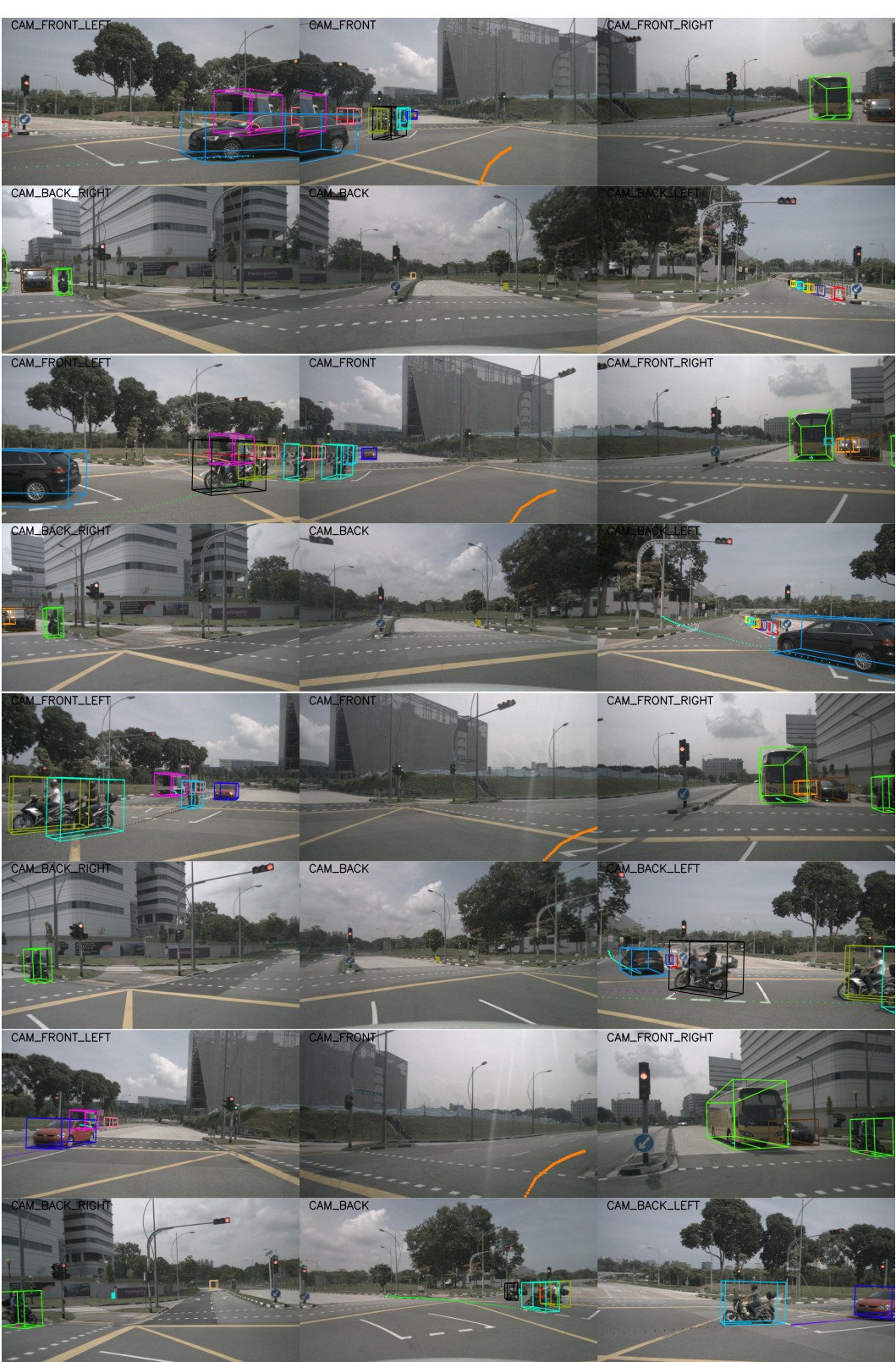

Figure 4: Qualitative results on confused road intersection scenario (multiple cross line on intersection).

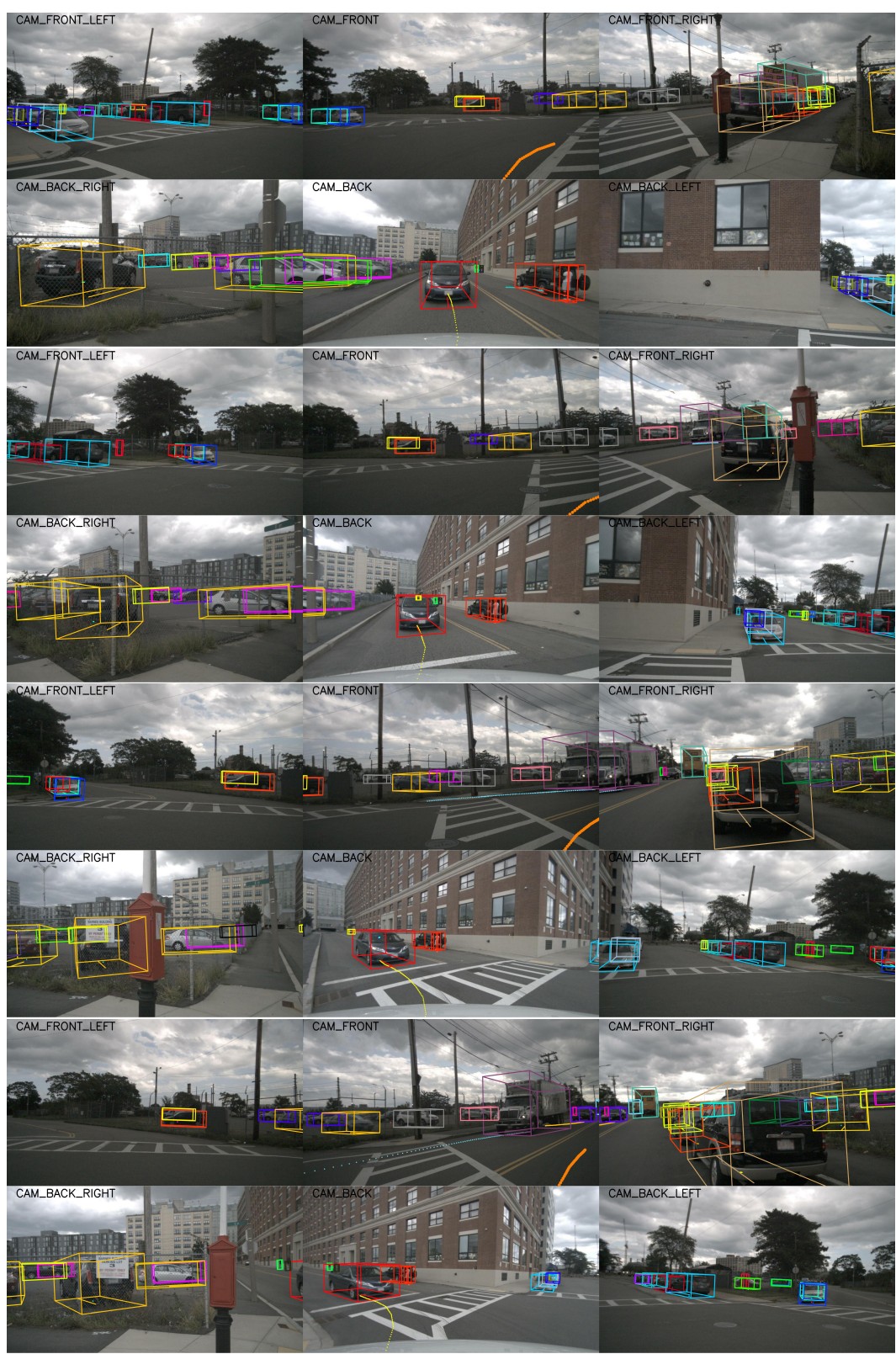

Figure 5: Qualitative results on turning scenario that surrounded by vehicles.

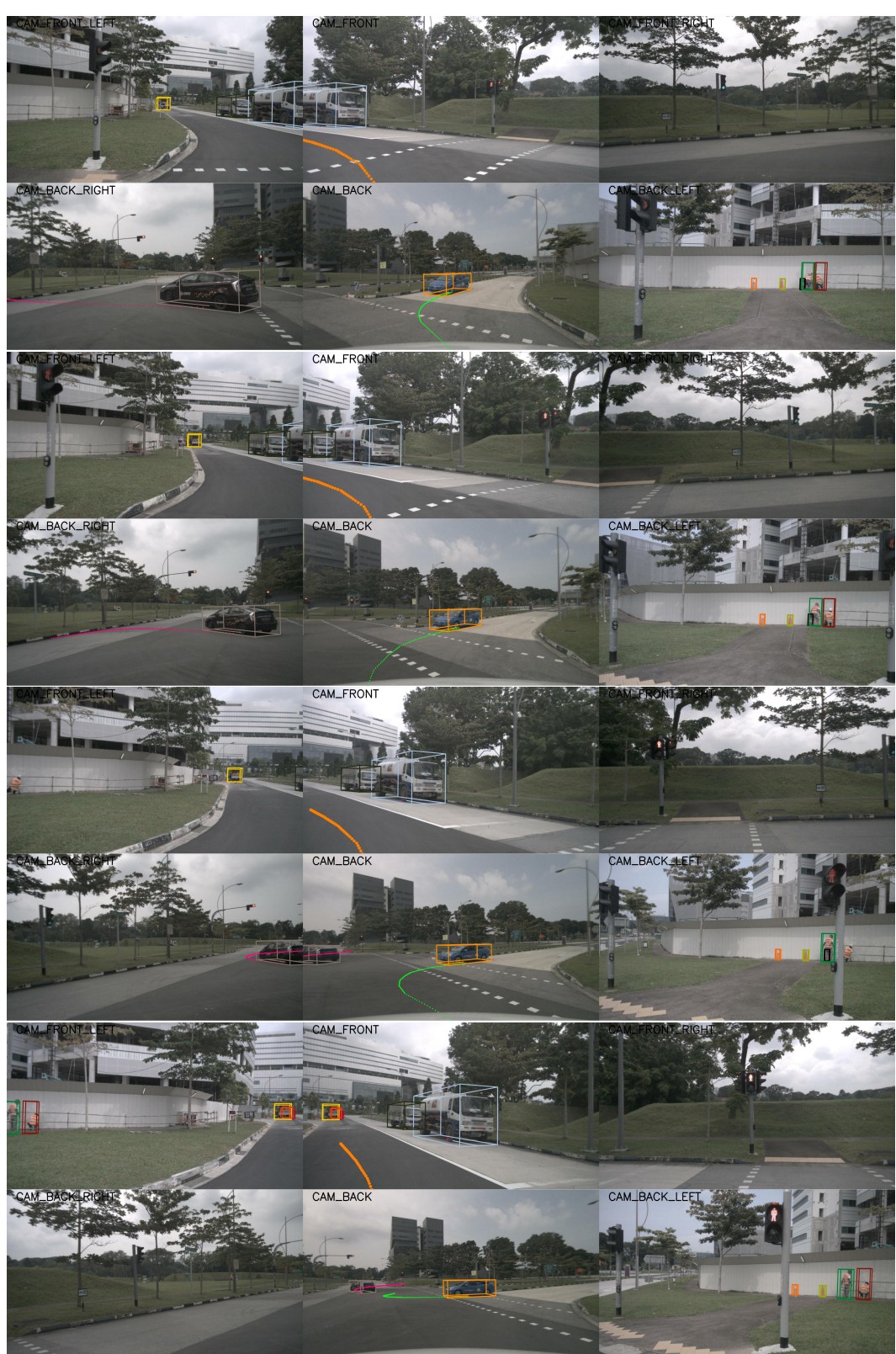

Figure 6: Failure case: Qualitative results of complex prediction trajectories.

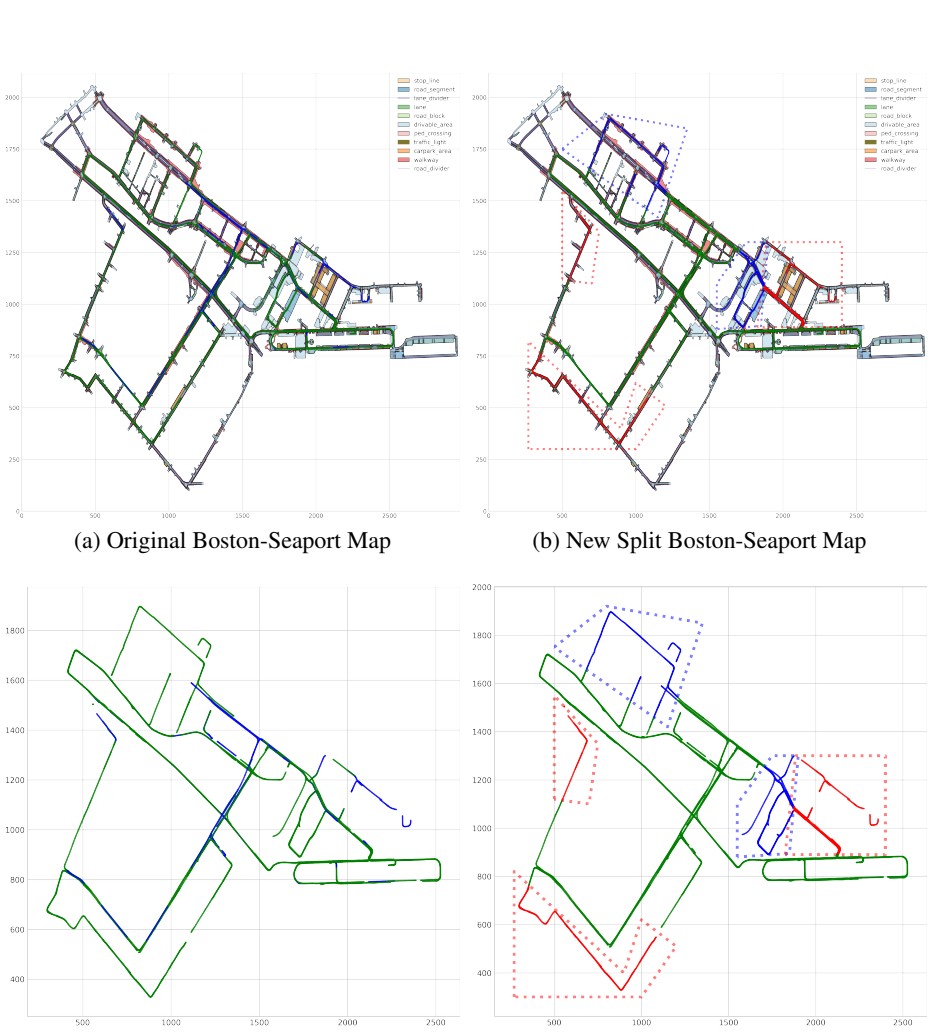

(a) Original Boston-Seaport Map    (b) New Split Boston-Seaport Map

(c) Original Boston-Seaport Data distribution    (d) New Split Boston-Seaport Data distribution

Figure 7: Comparison of original and new split datasets on the Boston Seaport map in nuScenes dataset. (a) and (c) are the original split visualization, (b) and (d) are the new split visualization. To ensure diversity in zone types within each set, regions from various parts of the city are included. Green: Train, Blue: Validation, Red: Test.

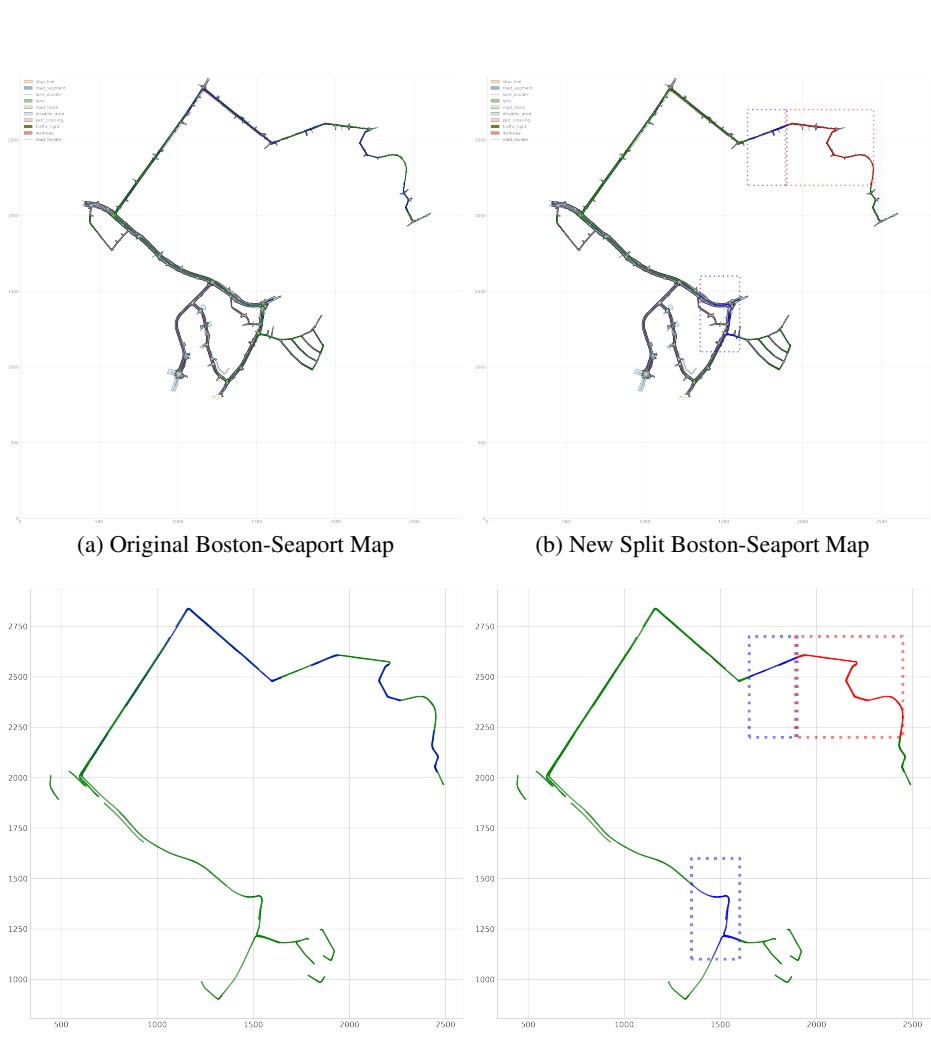

(a) Original Boston-Seaport Map

(b) New Split Boston-Seaport Map

(c) Original Boston-Seaport Data distribution

(d) New Split Boston-Seaport Data distribution

Figure 8: Comparison of original and new split datasets on the Singapore Hollandvillage map in nuScenes dataset. (a) and (c) are the original split visualization, (b) and (d) are the new split visualization. To ensure diversity in zone types within each set, regions from various parts of the city are included. Green: Train, Blue: Validation, Red: Test.

