# OpenReview forum: "AT-Drive: Exploiting Adversarial Transfer for End-to-end Autonomous Driving"
_ICLR.cc/2026/Conference — ICLR 2026 Conference Withdrawn Submission_

### Official Review · Reviewer_oFBW · 2025-10-26

**Soundness:** 3
**Presentation:** 3
**Contribution:** 3
**Rating:** 6
**Confidence:** 3

**Summary:**

* This paper addresses the challenge that e2e autonomous driving models trained on real-world data lack exposure to rare, hard cases, while models trained on simulation data that can focus on these rare scenarios suffer from a sim-to-real gap.
* The authors propose AT-Drive, an e2e framework that uses adversarial learning to transfer imitation driving capabilities learned from simulation to a real-world model.
* The system consists of two identical models and two discriminators. The discriminators (one for perception, one for motion) are trained to distinguish between perception and motion features respectively from the simulation model and the real-world model.
* The framework uses a two step adversarial training process. First, only the discriminators are trained to learn the data discrepancies. Second, the discriminators are frozen and only the real-world model is trained (with both real-world and simulation data) to produce features that fool the discriminators, thereby absorbing the simulation model's knowledge of hard cases.
* For evaluation, the authors created a new split of the nuScenes dataset: Simple scenarios for training and reserving challenging scenarios for validation. On this new split, AT-Drive achieves state-of-the-art performance, significantly reducing collision rates compared to baseline methods.

**Strengths:**

* The paper tackles the important problem of improving e2e autonomous vehicle models by leveraging simulation data for long-tail scenarios.
* The authors demonstrate that their methods work well (strong improvements in ADE and collision rates) compared to various baselines on both the nuScenes dataset and on a new newScenes dataset split. The ablation study further demonstrates that the proposed training regime is responsible for the wins.
* The authors create a new nuScenes dataset split of easy (train) and hard (eval) segments.

**Weaknesses:**

* The training seems rather brittle and the authors acknowledge it in the appendix and detail their choices for tuning gradient clipping, etc. It would be great to investigate this issue further and work on more principled techniques for stabilization.
* The training approach (two models + two discriminators + two stages) is rather expensive. Can this be improved (e.g. shared backbone, distillation, etc.)?
* In the conducted experiments, the difference between the simulation and real-world datasets is not just the simulation / real-world data source but also the distribution of examples (particularly regarding their difficulty). During evaluation, the dataset is focused on “medum-hard” examples but these don’t contain accidents, etc., which is contained in the simulation dataset. It would be good to also evaluate whether the model learns to handle these scenarios in the real world well, or whether it only got better in general on the “medium’hard” scenarios.

**Questions:**

* What’s the simulation / real-world data mixture used during the different training stages? Please provide these details in the paper.
* What are the discriminator losses used during the different training stages? Particularly, when using simulation vs. real world data in Stage 2 in the real world model, should simulation data be classified by the discriminators as real world data?

---

### Official Review · Reviewer_GrrS · 2025-10-27

**Soundness:** 2
**Presentation:** 2
**Contribution:** 2
**Rating:** 2
**Confidence:** 5

**Summary:**

This paper proposes to use adversarial training to transfer the knowledge learned from simulation data into the real-world domain. The experiment is conducted on a newly partitioned nuScenes dataset. It shows the proposed training method achieves SOTA performance in terms of several open-loop metrics.

**Strengths:**

The idea of transferring knowledge learned from one domain to another is novel. The simulated dangerous scenarios can be a good supplement to the real-world nominal driving data.

**Weaknesses:**

The main weakness of this paper is the poor evaluation. It should be evaluated in the widely used closed-loop setting and also compared to other SOTA methods on the leaderboard, such as Navsim 2.0. It will be considered as a sound solution only if it indeed helps reduce collisions and maintain the nominal driving performance, and gets an SOTA score on these benchmarks. The current evaluation is limited to a customized setting and open-loop metrics only, which makes it unclear how useful the proposed method is.

On the paper writing side, more figures can be used to illustrate the idea. The only method figure is very hard to read, with a lot of useless details. In the experimental section, it is better to include some visualizations on how the vehicle's behavior changes after doing the transfer. By having a comprehensive evaluation and visualization, I believe a lot of redundant content restating the idea of GAN in the method section can be removed as well.

Minor point:
- Use \citep instead of \cite sometimes

**Questions:**

N/A

---

### Official Review · Reviewer_SF2K · 2025-10-31

**Soundness:** 3
**Presentation:** 3
**Contribution:** 3
**Rating:** 6
**Confidence:** 4

**Summary:**

This paper introduces AT-Drive, an adversarial transfer framework for end-to-end autonomous driving that leverages both simulation and real-world data.
The key idea is to pre-train two identical models, one on simulation data that covers rare and risky scenarios and one on real-world data that captures naturalistic scenes, and then use two discriminators for perception and motion to align their feature distributions through adversarial learning.
A two-step back-propagation strategy ensures stable training by updating the discriminators and real-world model alternately, allowing the real-world model to absorb robustness and recovery skills learned in simulation.
Experiments on a newly partitioned nuScenes split and the NAVSIM closed-loop benchmark demonstrate large improvements over prior end-to-end methods such as UniAD and SparseDrive.

**Strengths:**

- Addresses one of the hardest gaps in autonomous driving, using simulated robustness to improve real-world reliability.

- Dual-discriminator structure separates perception and motion alignment in a principled way.

- Demonstrates consistent improvement on nuScenes and closed-loop tests with reasonable ablations.

- The training procedure is modular and transferable to other multimodal driving frameworks.

- Achieves 25 percent lower trajectory error and 60 percent fewer collisions compared to state-of-the-art methods.

**Weaknesses:**

- The core idea extends standard GAN-based domain adaptation, and the claimed unique back-propagation is similar to gradient reversal.

- The paper does not explain why two discriminators outperform a single one.

- Real-world deployment is demonstrated through datasets only.

- It is unclear if adversarial transfer causes degradation on easy or in-distribution scenes.

**Questions:**

Are the discriminators applied on latent features or on motion trajectories?

What prevents the real-world model from inheriting simulation biases or unrealistic motion priors?

---

### Official Review · Reviewer_77Jr · 2025-11-03

**Soundness:** 3
**Presentation:** 2
**Contribution:** 2
**Rating:** 2
**Confidence:** 4

**Summary:**

The authors present an end-to-end driving approach which utilizes data from both simulation and real world domains. Domain-agnostic perception and downstream motion features are optimized via two separate discriminator objectives. The authors’ setup enables a real world end-to-end driving policy to leverage simulated data, especially for rarer scenarios which may be difficult to obtain in the real world. To validate their method, they introduce a variant of nuScenes in which the training dataset involves simple, mostly-straight behavior, and the evaluation set involves more challenging behavior.

**Strengths:**

- The problem is well-motivated and the proposed approach is interesting.
- Authors propose solutions to rectify limitations of evaluation and training instability, given their unique goal of trying to generalize increasingly complex maneuvers across the domain gap.
- Experiments in open-loop planning show strong advantage over recent end-to-end driving approaches, given the same training setup.

**Weaknesses:**

- I’m not entirely sure that the backpropagation strategy proposed is novel. From my understanding, the authors improve adversarial training stability by alternating updates to the discriminators and the policy networks, to avoid conflicting gradients. The original GAN paper also alternatives gradient updates to the discriminator and the generator (https://arxiv.org/pdf/1406.2661), so I’m not exactly sure how the proposed backpropagation strategy differs other than in task (swap generator for driving policy).
- I find section 4.1 and 4.2 to be quite lengthy given that it seems primarily to 1) set up the adversarial problem and 2) contextualize the adversarial objective. For example, I believe that these two sections can be truncated to define the objective, with the full text being referenced in the appendix. Section 4.1 seems to describe a very generic domain adaptation setup.
- Lack of qualitative results. The problem which authors are trying to address is very interesting, but also difficult to evaluate with strictly aggregated metrics (Table 1). Some qualitative visualizations of learned behavior would be helpful to show the performance gap from Table 1.
- Scenario-specific evaluation. How do authors evaluate whether successful simulation-only actions were transferred to the real-world policy? Evaluation across simulator-only scenario types within the real-world dataset would be very useful. For example, performance in real-world unprotected turn behavior (only present in simulator data at train-time) would show skill transfer across the sim2real gap.

**Questions:**

- L457: Extraneous bottom line. This table could be better formatted.
- Are the losses in Eq 12-14 by any particular scaling coefficients? If so, please denote that there are scaling coefficients and provide the values authors found to lead to stable convergence in appendices — this would be really helpful for reproducibility, especially since authors mentioned convergence in practice being difficult.
- L423-425. I am confused by this particular setup. Are the authors generating traffic-violation behavior in the simulator training set for the ego policy to _avoid_, or _replicate_? It seems that the learned policy is trained in open-loop; in other words, I’m not sure that collision avoidance behavior can be captured robustly by including this data, unless the expert policy in simulation demonstrates robust avoidance behavior. Can authors clarify this section?

---

### Note · Authors · 2025-11-14

I have read and agree with the venue's withdrawal policy on behalf of myself and my co-authors.